# Radiation Defects in Heterostructures 3C-SiC/4H-SiC

**A.A. Lebedev [1,2,]\*** , **G.A. Oganesyan [1]**, **V.V. Kozlovski [3]**, **I.A. Eliseyev [1]** and **P.V. Bulat [4]**

[1] Ioffe Institute, Politekhnicheskaya 26, St. Petersburg 194021, Russia;
Gagik.Oganesyan@mail.ioffe.ru (G.A.O.); Valery.Davydov@mail.ioffe.ru (I.A.E.)

[2] Ioffe Institute, St. Petersburg State Electrotechnical University LETI named after V.I. Ulianov (Lenin),
St. Petersburg 197376, Russia

[3] Peter the Great St. Petersburg State Polytechnic University, St. Petersburg 195251, Russia;
Kozlovski@tuexph.stu.neva.ru

[4] Laboratory of Mechanic and Energy Systems, ITMO University, St. Petersburg 197101, Russia;
pavelbulat@mail.ru

\* Correspondence: shura.lebe@mail.ioffe.ru; Tel.: +7-812-2927125; Fax: +7-812-297017

**Abstract:** The effect of 8 MeV proton irradiation on n-3*C*-SiC epitaxial layers grown by sublimation on semi-insulating 4*H*-SiC substrates has been studied. Changes in sample parameters were recorded using the Hall-effect method and judged from photoluminescence spectra. It was found that the carrier removal rate (Vd) in 3C-SiC is ~100 cm$^{-1}$, which is close to Vd in 4*H*-SiC. Compared with 4*H* and 6*H* silicon carbide, no significant increase in the intensity of the so-called defect-related photoluminescence was observed. An assumption is made that radiation-induced compensation processes in 3*C*-SiC are affected by structural defects (twin boundaries), which are always present in epitaxial cubic silicon carbide layers grown on substrates of the hexagonal polytypes.

**Keywords:** sublimation epitaxy; 3C-SiC; proton irradiation; structural defects; photoluminescence; Hall effect; Raman spectroscopy

---

## 1. Introduction

It is known that silicon carbide (SiC) is a promising material for development of high-temperature, high-power, and high-frequency devices. Among the multitude of SiC polytypes, 3*C*-SiC is distinguished by the highest electron mobility, which is one of the most important characteristics of a material for the manufacture of devices. In addition, 3*C*-SiC has a cubic lattice and, by virtue of its symmetry, isotropic physical properties, in contrast to other SiC polytypes. No industrial growth technology of 3*C*-SiC substrates has been developed so far, and, therefore, 3*C*-SiC layers are most frequently grown on silicon substrates. Nevertheless, these layers have a poor crystal perfection because of the large lattice mismatch between Si and SiC (~20%). Therefore, it is of interest to examine the growth of 3*C*-SiC on the available substrates of other polytypes (4*H*-SiC, 6*H*-SiC), the lattice constants of which differ in the third decimal place. The main difficulty involves the numerous twinning regions that appear in the initial stage of the epitaxial growth.

It has been shown previously that n- and p-type 3*C*-SiC epitaxial layers with thickness of 3–5 μm can be formed on SiC substrates of hexagonal polytypes (4*H*-SiC) by the method of sublimation in a vacuum [1–4]. The goal of the present study was to examine the radiation hardness of the epitaxial layers.

## 2. Experimental

In this study, epitaxial 3*C*-SiC layers were grown by sublimation epitaxy in a vacuum. Semi-insulating substrates of the 4*H* polytype served as substrates. The growth was performed

on polar C $(000\bar{1})$ and Si (0001) faces of a substrate. The growth temperature was 1950–2000 °C; growth duration, 10 min; and area of a thus grown 3*C*-SiC layer, ~1 cm$^2$. The carrier concentration in the 3*C*-SiC epilayer was $6.5 \times 10^{17}$ cm$^{-3}$, and the thickness of the layer was 10 μm. Commercial finely grained silicon carbide powder with grain diameter of 10–20 μm was used as the source. When a cubic epitaxial layer nucleates simultaneously at different points of a hexagonal substrate, there are two kinds of orientation of 3*C*-SiC nuclei along the growth surface, which differ in being turned by 60° with respect to each other (twinning structure) [1]. Defective regions (double-position boundaries, DPBs) are formed at boundaries between the 3*C*-SiC twins.

To confirm the polytype of the epitaxial layer in 3*C*-SiC/4*H*-SiC heterostructure, Raman spectroscopy was used. Polarized spectra were measured at room temperature under the 532 nm laser light excitation in the $z(xx)\bar{z}$ and $z(yx)\bar{z}$ scattering configurations. Here $z$ and $\bar{z}$ are perpendicular to the (0001) plane of the substrate, while $x$ and $y$ are mutually orthogonal and lie in the plane perpendicularly to the $z$ direction.

The proton irradiation was performed on an MGTs-20 cyclotron (NIIEFA, St.Petersburg, Russia) with 8 MeV protons and irradiation doses (D) in the range from $3.0 \times 10^{14}$ to $6.0 \times 10^{15}$ cm$^{-2}$.

Ohmic contacts for Hall-effect measurements were fabricated on the epitaxial layers. For this purpose, 0.3-μm-thick nickel (Ni) layers were deposited above 30-nm-thick titanium (Ti) films. Furthermore, the contacts were annealed at 850 °C in a vacuum for 90 s.

The photoluminescence (PL) was excited with a nitrogen laser at a wavelength of 337.1 nm. The laser had the following parameters: pulse power 2 kW, pulse width 10 ns, and pulse repetition rate 100 Hz. The pumping power density was ~50 kW/cm$^2$. The PL spectra were measured at liquid-nitrogen temperature (77 K).

## 3. Results and Discussion

Figure 1 shows polarized Raman spectra obtained in the nearly backscattering geometry for the heterostructure before the proton irradiation. It is seen that the spectrum contains signals from both the epilayer and the substrate. The symmetry and the Raman frequencies of the phonon modes presented in Figure 1 are in a very good agreement with those reported for the 4*H*-SiC and 3*C*-SiC polytypes [5]. The peaks at 204 cm$^{-1}$, 610 cm$^{-1}$, 777 cm$^{-1}$, and 964.5 cm$^{-1}$ belong to the 4*H*-SiC folded phonon modes. The peaks at 796.5 cm$^{-1}$ and 973.8 cm$^{-1}$ can be attributed to the *TO* and *LO* of 3*C*-SiC optical phonons, respectively.

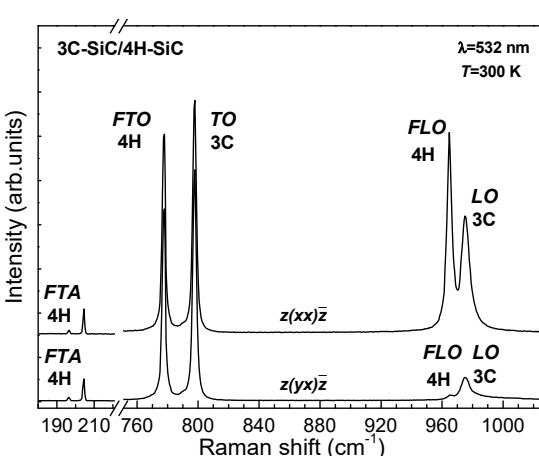

**Figure 1.** Polarized Raman spectra of 3*C*-SiC/4*H*-SiC heterostructure before the proton irradiation.

The carrier concentration and mobility were found from the Hall effect after each irradiation dose. Figure 2 shows how the carrier concentration depends on irradiation dose for the 3*C*-SiC/4*H*-SiC heterostructure. It can be seen that the carrier concentration linearly decreases with an increasing irradiation dose. This demonstrates that the compensation mechanism in 3*C*-SiC films is the same as

that for the hexagonal silicon carbide polytypes: transition of electrons from donor levels to levels of the acceptor-type radiation defects being formed [6].

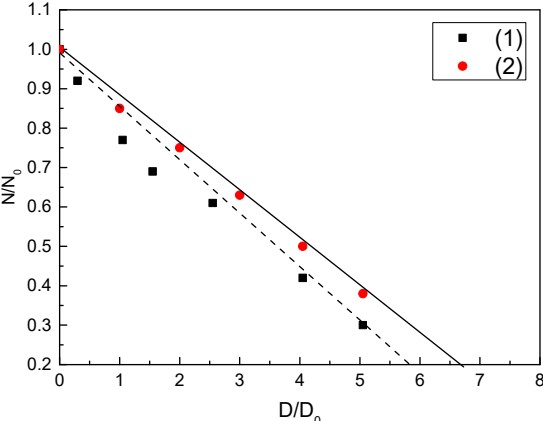

**Figure 2.** Normalized dependences of the electron concentration on irradiation dose: (1) 3*C*-SiC/6*H*-SiC structures, $D_0 = 1 \times 10^{12}$ cm$^{-2}$, $N_0 = 8 \times 10^{14}$ cm$^{-3}$, (2) 4*H*-SiC epitaxial layers, $D_0 = 1 \times 10^{15}$ cm$^{-2}$, $N_0 = 6.5 \times 10^{17}$ cm$^{-3}$.

The value of Vd was calculated according to the formula: $Vd = (N0–N1)/\Delta D$, where $N0$ is the electron concentration in the epitaxial layer before *the* irradiation, $N1$ is the concentration in the epitaxial layer after the irradiation, and $\Delta D$ is the irradiation dose. It is somewhat unexpected that the carrier removal rate in 3*C*-SiC films (Vd) is ~100 cm$^{-1}$, similarly to 4*H*-SiC, although the energy gap width in 3*C*-SiC is 0.9 eV smaller than that in 4*H*-SiC. Possibly, the radiation defects partly go to drains formed by twin boundaries and are not involved in the compensation process.

The results obtained in a study of the electron mobility are presented in Figure 3. It can be seen in the figure that the electron mobility nearly linearly decreases with increasing irradiation dose. Apparently, this is due to the rise in the concentration of charged radiation defects, which are additional carrier-scattering centers. Interestingly, the mobility at room temperature twice decreases after the maximum dose.

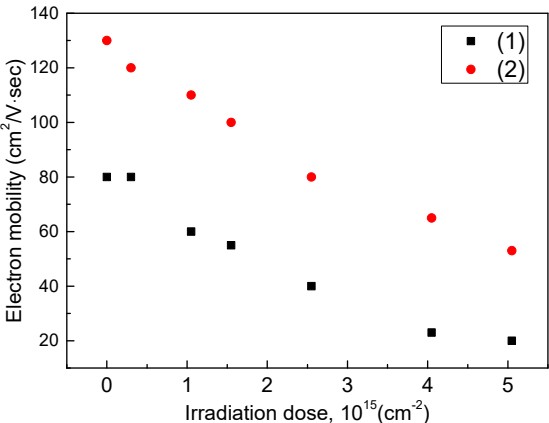

**Figure 3.** Electron mobility in 3*C*-SiC layers vs. proton irradiation dose: (1) 77 K and (2) 300 K.

At the same time, the mobility becomes more than three times lower at liquid-nitrogen temperature. Apparently, this occurs because the scattering on phonons plays a more important part at room temperature, compared with the scattering on impurity atoms and defects.

It is known that longitudinal optical modes in polar semiconductors interact via macroscopic field with collective excitations of free carriers, thus forming LO-phonon-plasmon coupled (LOPC) modes. The frequencies of LOPC modes are strongly dependent on the doping level. The carrier concentration

is deduced by analyzing the line shape and the frequency of the LOPC mode in Raman scattering [7]. Figure 4 shows as an example the transformation of the Raman spectrum of the heterostructure in the range of the LOPC modes after the irradiation with protons at an irradiation dose of $6.0 \times 10^{15}$ cm$^{-2}$. The spectra demonstrate a low-frequency shift and a narrowing of the LOPC mode for the irradiated 3C-SiC epilayer. The carrier density $n$ was determined by fitting the theoretical curve equation to the line shape of LOPC modes in Figure 4, as described in reference [8]. The estimated values of $n$ are given in Figure 4. It can be seen that the carrier concentration in the 3C-SiC epilayer decreased significantly after the irradiation, which agrees well with the results of the Hall measurements.

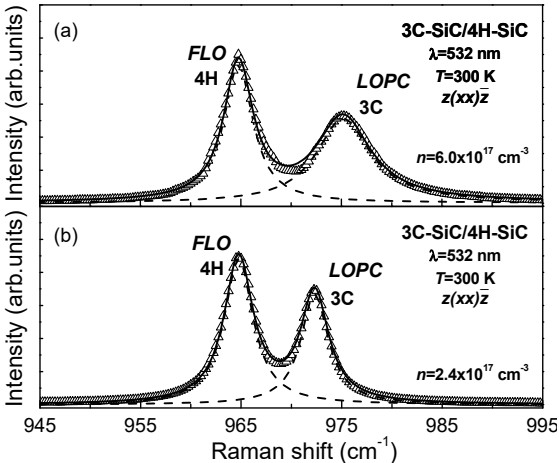

**Figure 4.** Polarized Raman spectra of the heterostructure in the range of the LOPC modes: (**a**) as grown, and (**b**) after the irradiation with 8 MeV protons at irradiation dose of $6.0 \times 10^{15}$ cm$^{-2}$.

The PL spectra of all the samples, both as-grown and after each irradiation dose, were examined (Figure 5). It is known that irradiation with electrons, protons, and various kinds of ions leads to an increase in the intensity of the so-called defect-related PL (DPL) in 4*H*- and 6*H*-SiC [9]. The DPL has also been observed upon ion irradiation in 3*C* single-crystals and heteroepitaxial films grown on silicon [10]. An analysis of the experimental results obtained in reference [11] demonstrated that the dependence of the DPL intensity on irradiation dose is well accounted for on the assumption that this PL is due to the donor-acceptor pair (DAP) recombination on the pair constituted by nitrogen and a radiation defect. The emission peak is shifted in various SiC polytypes in relation to the energy gap width of a polytype. In 3*C*-SiC polytype, the peak in the DPL spectrum lies at energies of 1.7–1.8 eV.

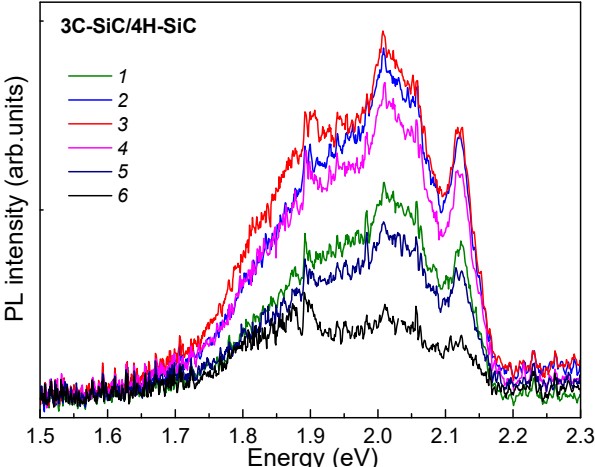

**Figure 5.** PL spectra of epitaxial 3C-SiC samples after various irradiation doses, $10^{15}$ cm$^{-2}$: (1) 0, (2) 1, (3) 2, (4) 3, (5) 4, (6) 5.

In the present study, no significant increase in the DPL intensity was observed after the irradiation. In our samples (3C-SiC films grown on hexagonal substrate) structural defects are concentrated on the boundaries of the 3C-SiC twins forming the defective regions (Double position boundaries—DPB).

In 3C-SiC single crystals and films grown on silicon, this DPB are absent. Therefore, we believe that the acceptor level associated with a structural defect, which is a component of the donor-acceptor pair, goes to drains formed by DPB. Therefore, the intensity of "defective" photoluminescence does not increase in our structures after irradiation. However, we think that this issue requires more detailed study and we plan to carry it out in our future work.

## 4. Conclusions

It was found that the compensation of cubic silicon carbide upon irradiation, which is similar to that of the hexagonal SiC, is due to the transition of electrons to deep acceptor-type radiation defects being formed. It was demonstrated that the 3*C*-SiC epitaxial layers grown in the study have approximately the same radiation hardness under irradiation with protons as that for 6*H*- and 4*H*-SiC. Nevertheless, the cubic silicon carbide polytype does not exhibit, in contrast to the hexagonal polytypes, any increase in the intensity of the defect-related photoluminescence. In the authors' opinion, the last two circumstances may be due to the removal of a part of the radiation defects to drains, which are twin boundaries in 3*C* films. This twin structure is a characteristic structural defect for epitaxial 3*C* layers grown on SiC substrates of hexagonal polytypes.

**Author Contributions:** C-V measurement and paper text preparation—A.A.L.; Hall effect measurement—G.A.O.; Samples irradiation—V.V.K.; Raman measuremet—I.A.E.; Photoluminescence measurement—P.V.B.

**Funding:** The study was financially supported by the Ministry of Education and Science of the Russian Federation (Agreement no. 14.575.21.0148, unique project number RFMEF157517X0148).

**Conflicts of Interest:** The authors declare no conflict of interest.

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
