# Peer review of "Radiation Defects in Heterostructures 3C-SiC/4H-SiC"

_crystals, doi:10.3390/cryst9020115_

Reviewer 1 Report

    The authors of the submitted manuscript investigate irradiation effect of a 8 MeV energy proton beam on a cubic silicon carbide 3C-SiC film grown on hexagonal 4H-SiC semi-insulating substrate. SiC is a wide band-gap semiconductor having outstanding properties (very inert chemically, biocompatible, interesting mechanical & high thermal transport properties, high carrier mobility, and resistance to radiation damages). So, within this scope, the investigation presented in this submitted manuscript is of special interest.

    The authors use the Hall effect and photoluminescence experimental techniques to probe the effect of protons on such a cubic 3C-SiC film grown compared to the behavior of hexagonal SiC. Both SiC polytypes appear to have about the same resistance to the proton beam irradiation. Most interestingly, contrary to the hexagonal SiC polytypes, the cubic 3C-SiC does not show any significant subsequent defect density increase as evidenced by the photoluminescence experiments. It is suggested that such a behavior could result from these defect to drains.

    So this work looks of interest to the community and therefore, is suitable for publication.

Author Response

First of all, we would like to thank the reviewer for the comments made.  We agree with the comments made. In order to improve the text, passed   it to a professional translator and made the necessary corrections.

Reviewer 2 Report

The paper presents an interesting study on the radiation defects in 3C-SiC/4H-SiC hetero-structure. The presentation of the results is clear with the exception of the PL results. In fact, the complex behaviour shown in figure 5 it is not explained in detail. Futhermore, in the text is reported that this behaviour is different with respect to the behaviour observed in the case of the 3C-SiC grown on silicon substrates. Probably a comparison of these literature spectra with the experimental ones in the figure could improve the clarity of the paper.

Author Response

We are grateful to the reviewer for the comments made.  We believe that the so-called “defect” photoluminescence does not appear  in our structures due to a different composition of structural defects.   When a cubic epitaxial layer grows on a hexagonal SiC, nucleation    begins simultaneously at different points (locations) of the hexagonal    substrate, the orientation of the 3C-SiC nuclei along the growth     surface has two types differing in reversal relative to each other     at 60º (twin structure). As a result, structural defects are     concentrated on the boundaries of the 3C-SiC twins forming the     defective regions (Double position boundaries - DPB).In 3C-SiC single crystals and films grown on silicon, this DPB are  absent. Therefore, we believe that the acceptor level associated with a  structural defect, which is a component of the donor – acceptor pair,   goes to drains formed by DPB. Therefore, the intensity of “defective”    photoluminescence does not increase in our structures after     irradiation. However, we also agree with the reviewer that this     issue requires more detailed study and we plan to carry it out in our     future work. We have done changes in text.

Reviewer 3 Report

 Given the proximity of the energy of the chemical bond of the silicon carbide polytypes, we should expect them to have close radiation hardness. The correspondence of charge carriers removal rates under irradiation with protons in this case is most likely due to the close of the Fermi level positions in the starting materials. 

Author Response

We would like to thank the reviewer for the comments made.  We agree with the comments made